# The Pitfalls in the Path of Probabilistic Inference in Forensic Entomology: A Review

**DOI:** 10.3390/insects12030240

**Published:** 2021-03-12

**Authors:** Gaétan Moreau

**Affiliations:** Département de biologie, Université de Moncton, Moncton, NB E1A 3E9, Canada; gaetan.moreau@umoncton.ca; Tel.: +1-506-8584975

**Keywords:** animal carcass, cadaver, decaying substrate, insect succession, postmortem interval, successional studies, vertebrate decomposition

## Abstract

**Simple Summary:**

Experimental studies in forensic entomology must follow a series of rules to generate accurate predictions about criminal cases. These rules are reviewed and some approaches that have been used to solve experimental problems in forensic entomology are presented. Finally, recommendations are provided to avoid the publication and possible use in court of forensic studies that fall short of experimental standards.

**Abstract:**

To bridge the gap between experimentation and the court of law, studies in forensic entomology and other forensic sciences have to comply with a set of experimental rules to generate probabilistic inference of quality. These rules are illustrated with successional studies of insects on a decomposing substrate as the main example. The approaches that have been used in the scientific literature to solve the issues associated with successional data are then reviewed. Lastly, some advice to scientific editors, reviewers and academic supervisors is provided to prevent the publication and eventual use in court of forensic studies using poor research methods and abusing statistical procedures

## 1. Forensic Entomology, an Inferential Science

Inference is defined as an operation by which one moves from one assertion that is considered true to another assertion by means of rules that make that second assertion equally true. Considering this, much of the research carried out in forensic entomology, as in other forensic sciences, is fundamentally inferential because experiments carried out on samples are often used to draw conclusions about the general population. For example, it is common to carry out studies using cadavers [1,2], whole animal carcasses [3,4] or animal tissues (reviewed in [5]) with the aim of later making extrapolations from experimental data to estimate a postmortem interval (PMI), a period of insect activity (PIA) or a post-colonization interval (PCI) in a criminal case. Reasoning from data to an individual case presents considerable challenges [6]. In addition, as the conditions of the experiment do not correspond in all points to those of the case, compliance with a set of rules is required to ensure the quality of the inference being drawn. These rules have been thoroughly described for over 100 years in treatises on experimental design and statistics [7,8,9,10], and this is where the problem lies: most practitioners in forensic sciences have limited statistical literacy. While several recent publications have sought to provide tools to forensic researchers wishing to improve the quality of methods, experimental design and statistics [11,12,13,14], forensic publications and conference presentations with poor research design and data analysis are still common. It is important to recognize that studies that fall short of experimental standards encourage false-positive findings, which can lead to wrongful convictions or exonerations. Incidentally, recent miscarriages of justice have often been attributable to flawed expert testimony [15].

What are the main methodological issues observed in the forensic entomology literature that make inference impossible? The foremost issue is a lack of compliance with Fisher’s four cornerstones of experimental design: randomization, replication, independence, and controls [8,9]. This has already been discussed in recent literature reviews [12,13,14] and although some progress has been made in the last nine years, the absence of treatment replication, also called simple pseudoreplication, is still rampant in forensic entomology. A common form of simple pseudoreplication is to use one replicate per treatment (e.g., one carcass per condition, a single growth chamber per temperature for a study on maggot development, a single field and forest to test for differences between habitats, collecting data for a single year, i.e., one spring, summer, fall and winter for a study on seasonal effects) and consider subsamples to be true replicates when interpreting the data. The second issue is an absence of inferential statistical tests permitting the generation of probabilistic inference. It has been argued elsewhere that probabilistic inference procedures are an appropriate approach for measuring uncertainty in forensic science [16,17]. However, forensic entomology remains a descriptive science where inferential statistical tests are the exception rather than the rule and where the uncertainty associated with indicators of PMI, PIA or PCI is seldom disclosed. A good example of a widespread practice which impedes probabilistic inference is the presentation of summary tables or histograms pooling the information of several cadavers/carcasses [12].

There are however cases where even with a well-planned study, the very nature of the experiment is binding and imposes some adjustments to allow for inference. This is particularly true for studies of insect succession on cadavers or vertebrate carcasses, probably the most complex type of experiment from which one would extract a probabilistic inference in forensic entomology. Below, I substantiate this assertion by laying out the methodological issues associated with successional data recovered during studies of succession on necromass. While forensic entomology is used as an example, most of the elements discussed here apply to other forensic sciences as well.

## 2. Issues Associated with Successional Data

Successional data are samples, measurements, statistics or any factual information recovered sequentially from a decaying substrate. In forensic entomology, a good example of successional data is the maggot sample recovered at different time intervals from a cadaver/carcass to describe fly ontogenic and morphological changes [18,19]. Another example of successional data is the insect sample [1,2] or the visual count of insects [20] recovered through time from a cadaver/carcass. Then again, samples of microorganisms/bacteria [21], fungus [22], volatile organic compounds [23], synovial fluid and vitreous humour [24], or taphonomical changes (i.e., body scores, decomposition stages [25,26,27]) recovered from decomposing necromass are also successional data. To be clear, most of the time, information used to estimate the postmortem interval is successional data. What many overlook is that generating inferences from these datasets represents a challenge because of their inherent properties, which are listed in Table 1 and described below.

### 2.1. Data Measured Repetitively from a Small Number of Sampling Units and Field Sites

Most of the time, forensic studies use few cadavers/carcasses as sampling units because of the challenge associated with obtaining them and the time required to sample them. In addition, studies are usually conducted in the same environment year after year due to limitations associated with body farms or the difficulty in obtaining and accessing different experimental sites [28]. This has several consequences other than considerations linked with carcass enrichment [29,30], the first one being that records of successional data will be interdependent. Statistically, interdependence means that successive samples are correlated. This interdependence can sometimes be noticed visually during an experiment; if a rare insect is observed on a given carcass on day 2 of the study, it is more likely that the same insect will be observed the next day on the same carcass than on another carcass. A second consequence is the development of an autoregressive covariance structure, which means that interdependence is higher in two adjacent time periods and systematically decreases as the distance between the time periods increases. Again, this can be detected during an experiment as species recovered from a carcass/cadaver sampled on two successive days will be more similar than those collected over a longer time interval, such as day 1 and day 6 of the study. Both conditions indicate that the variance is not randomly distributed in the data, contravening a fundamental assumption of several statistical tests. A third consequence of having a small number of sampling units is that analyses will have limited statistical power. Statistical power is the likelihood that a hypothesis test will detect an effect if there is one. It depends on sample size and effect size, the latter being a quantitative measure of the magnitude of the experimental effect. This means that if a small difference exists between two sets of conditions (i.e., a small effect size), a large number of samples is necessary for this difference to be significant (for an example of insufficient sample size as revealed by power calculation in forensic entomology, see [13]). Conversely, if the effect size is large, few samples are needed to detect a difference between two sets of conditions (for an example, see [5]). A fourth consequence of having a small number of sampling units is that the study has a low internal validity because there is a greater chance that bias or random effects will have consequences on a significant portion of the sample. Internal validity describes the extent to which a study establishes a trustworthy cause-and-effect relationship and is not influenced by other factors. A low internal validity means that our confidence that manipulated variables caused a change in measured variables is compromised. In experiments with few carcasses/cadavers, it is much more difficult to detect bias and ascertain whether the data collected are valid and only affected by the variables under the control of the researcher than in larger studies.

### 2.2. Non-Linearity

Generally, successional data exhibit non-linear trends. Instead of increasing or decreasing linearly (first-order regression) or quadratically (second-order regression), variables often exhibit complex trends that can only be fitted with high-order polynomial regressions. For example, insect occurrence or abundance on cadavers/carcasses is bell-shaped or multimodal, with few insects documented at the beginning and end of succession, and many insects documented at an intermediate time interval [20,31]. High-order polynomial models are difficult to translate into biological terms, and their adjustment can lead to statistical model overfitting. Overfitting means that an overly complex model has been produced which fits the underlying relationship between the study variables as well as the noise unique to each sample. As a result, overfitted models produce misleading coefficients and although they generally perform perfectly with the experimental data, they are likely to perform poorly when other data (such as from a criminal case) are used [32]. In addition, when successional records of insects or residuals of analyses on successional records are plotted using frequency distribution histograms, a normal or Gaussian distribution is seldom obtained. Instead, distributions of forensic data tend to conform to the binomial distribution, the Poisson distribution or the quasi-Poisson distribution. The Poisson distribution is one of the best statistical distributions to model the number of times an event occurs in an interval of time or space, such as the number of times that *Phormia regina* (Diptera: Calliphoridae) is observed each day during succession on a given carcass in my study area. The first consequence of the two conditions identified above is that statistical models developed without considering non-linearity and non-Gaussian data distribution tend to have a poor fit. A second consequence is that approximations provided by the central limit theorem are likely to be inadequate, which signifies that issues linked to the lack of normality cannot be disregarded. A third consequence is that, usually, transformations applied to push the data closer to a Gaussian distribution perform poorly. A final consequence is that classical linear models (e.g., analysis of variance, regression) are often ineffective with such data.

### 2.3. Datasets with a Relatively Large Proportion of Unexplained Variance

Statistical tests can generally separate variance into two components: the explained and unexplained variance. The explained variance is the variance associated with the study variables whose influence is being studied such as the PMI and postmortem accumulation of degree-days in successional studies of insects. The unexplained variance is any residual variance associated with random variance that occurs because experimental units always exhibit some differences, even in homogeneous conditions, as well as variance associated with unknown variables, called systematic variance. Systematic variance is non-random variance due to factors not manipulated or measured during the study. In forensic entomology, systematic variance can overly dominate the explained variance because insect occurrence is influenced by meteorology, animal behavior, life histories, microhabitat, etc. This means that study variables often poorly explain the factors responsible for effects, and that the study has little external validity. External validity refers to the extent to which research findings can be generalized. A low external validity limits the ability to draw inferences from experiments and generalize the findings. This implies that it is incorrect to use the results of the study to explain a given situation occurring elsewhere, including a case study. For example, if only 15% of the variance is accounted for by a model developed in a laboratory study to predict the PIA from the size of maggots, this model is unusable in a forensic case because it has no predictive power.

### 2.4. Data Affected by Temporal and Spatial Effects

As a rule, all insect successional studies carried out in the field are inevitably affected by temporal and spatial effects. Temporal effects occur because the same experimental units (i.e., carcasses/cadavers) are sampled over and over during most forensic studies. In addition to sometimes causing oversampling problems [33], this leads to an interrelation of the samples over time. Moreover, repeated samples cannot be considered random because a sample taken at time 1 is necessarily collected before a sample at time 2. Such a dataset is unlikely to be stationary in the sense that the mean, variance, autocorrelation and other statistical properties are all constant over time, contravening another assumption of several statistical tests. Spatial effects occur because field conditions are often heterogeneous and because the placement of sampling units can create interdependence between units, thereby causing systematic spatial variation that results in observable data clusters. Cadavers/carcasses that are closely spaced can become linked by dispersing organisms that synchronize their dynamics [34]. For example, in a recent study [20], we noticed that less competitive carrion beetles such as *Necrophila americana* (Coleoptera: Silphidae) were kept at bay from carcasses by more competitive species such as *Necrodes surinamensis* (Coleoptera: Silphidae). The former species thus moved to less interesting habitat patches, producing a spatial structure explaining the results of the study. Overlooking or ignoring these spatial patterns, when they are present, could lead to false conclusions about cause-and-effect relationships [35]. This issue is particularly problematic for studies conducted in body farms or small field sites where cadavers/carcasses cannot be placed far apart.

### 2.5. Datasets Including Many Independent and Dependent Variables

Often, in forensic studies involving cadavers/carcasses, a large amount of data is collected from each experimental unit (i.e., cadavers/carcasses). For example, several studies record daily the presence of different taxa, the occurrence of egg masses, the size of maggots, the temperature of the cadaver, the stage of decomposition of the cadaver, the postmortem accumulation of degree days, etc. However, the scientific literature recommends that the number of variables remain low compared to the number of observations, because the chance of finding significant but biologically irrelevant relationships between the dependent and independent variables increases with the number of variables [36]. Ignoring this recommendation is likely to result in autocorrelation between processes as well as autocorrelation between response variables. When autocorrelation exists between two variables, this indicates that one of the study variables is a duplicate of the other variable more or less shifted in time or in space. For example, during the summer in my study area, the postmortem accumulation of degree days and the PMI are always autocorrelated because the temperature is relatively constant. Without proper statistical adjustment to deal with autocorrelation, multicollinearity, overfitting (discussed in Section 2.2) and Alpha inflation are inevitable. Multicollinearity is a process whereby dependence among the explanatory variables is strong enough that one explanatory variable can be accurately predicted from the others. Thus, the collinear explanatory variables contain the same information about the dependent variables. A model developed using collinear explanatory variables yields highly volatile parameter estimates with large standard errors [37]. With such a model, a small change in the data can result in a large change or even a change of sign in parameter estimates. Consequently, little confidence can be placed in a model affected by multicollinearity. Alpha inflation is also known as familywise error, experimentwise error or cumulative Type I error. Alpha (or α) refers to the significance level. In most experimental sciences, α = 0.05, which implies that you have a 5% chance of incorrectly rejecting the true null hypothesis when you perform a statistical test (i.e., you have a 5% chance of detecting an effect when there is none). As more tests are conducted on the same dataset, the likelihood of obtaining at least one erroneous significant result increases. For example, if the response to a given treatment (e.g., shading) of 20 blowfly species sampled in an experiment is analyzed individually with α = 0.05, there is 64.1% chance that one or several analyses of species responses will be significant due to chance. This number is obtained by the following calculation:α^′^ = 1 ‒ (1 ‒ 0.05)^20^ = 0.641(1)

In short, multicollinearity, overfitting and Alpha inflation imply that the interpretation of the results will be too liberal, that statistical inferences will be less reliable and that the whole study will potentially be misleading.

## 3. Possible Remedies to the Issues Associated with Successional Data

To address the problems discussed above, I list in Table 2 and review below some approaches that have been used to deal with these issues in forensic entomology and in studies of insects in other degradative systems (i.e., dung, dead wood).

### 3.1. How to Solve Problems Related to Low Statistical Power as well as to Low Internal and External Validity

These problems have a single cure: increase the sample size. If this is impossible because sampling is time-consuming, review the protocol to find out ways to speed up its implementation. Another method to increase external validity is to repeat the study in a variety of locations, times, and conditions. This may appear paradoxical because generally, one seeks to decrease rather than increase natural variability in experimental studies. However, the purpose of forensic science differs from process-driven science as the aim is generally not to conclude about the effects of a treatment, but rather to use the study results for the interpretation of data from a criminal case that occurred under different conditions than the study. A good example of the two precepts discussed above is given by Horgan [38] who studied dung beetles attracted to decomposing cow dung in 16 widely separated locations in the contrasting pasture landscapes of El Salvador and Atlantic Nicaragua. Because of all the variability that was consciously integrated into the design of this study, the observed trends are strong and easily transposable to other environments. For a similar example using saproxylic beetles, see [39].

### 3.2. How to Solve Problems Related to Interdependence between Records, Auto-Regressive Covariance Structure, Non-Linear Effects and Non-Gaussian Distributions

The solution to these problems is straightforward: use generalized linear models (GLMs), generalized linear mixed models (GLMMs), generalized additive models (GAMs) or generalized additive mixed models (GAMMs). Basically, these models work as analyses of variance, analyses of covariance or linear regressions but use non-linear link functions to allow for responses with nonlinear distributions such as Poisson, binomial, gamma, etc. These models can also account for data interdependence and for the autoregressive structure of the data. Moreover, additive models use non-parametric smoother functions to fit models with fewer assumptions. As the description of statistical models is not one of the objectives of this text, I take this opportunity instead to encourage readers to consult examples of the application of these models in recent forensic literature. For examples of GLMs/GLMMs, see [26,31,40,41]. For examples of GAMs/GAMMs, see [5,39,42,43].

### 3.3. How to Solve Problems Related to Autocorrelation, Multicollinearity, Overfitting and Alpha Inflation

The solution to these problems is forthright: use multivariate statistics. Multivariate statistics encompass all approaches that allow for the simultaneous analysis of several response variables. They can be grouped into different categories such as the descriptive or correlative methods (e.g., ordinations, canonical correlations and clustering methods), or the explanatory and predictive methods (e.g., regression trees, multivariate analysis of variance (MANOVA), discriminant analysis, random forests). These methods are useful to deal with autocorrelation, the choice of a method depending on the objectives of the study, the experimental design and the analyst’s preferences. When multicollinearity and overfitting are an issue, multivariate statistics can help with variable selection to reduce dimensionality and allow for further exploration and analysis of the data [44]. As mentioned above, the description of statistical models is not one of the objectives of this text. Thus, I encourage readers to consult examples of the application of these models in recent studies in forensics or other degradative systems. For examples of ordination and canonical correlations in a forensic context, see [40,45,46,47]. For examples of multivariate regression trees, see studies on saproxylic beetles such as [48,49]. For examples of discriminant analysis, see studies on saproxylic beetles such as [50] or on dung beetles such as [51]. For examples of MANOVAs in a forensic context, see [33,40,52,53].

### 3.4. How to Solve Problems Related to Having Data Interrelated in Time or Space

If the objective of the study is to analyze only the temporal trends of a dataset, time series analysis such as used by Andow and Kiritani [54] in a study of saproxylic beetles is appropriate. If variables other than time need to be included in the model, then the GLM, GLMM, GAM and GAMM approaches suggested above are adequate (for an example, see [5,20]). On the other hand, potential spatial relationships require the use of spatial statistics that specifically describe and model localized or geo-referenced data. To date, forensic entomologists have not ventured into spatial statistics, but this is bound to change as studies encompassing several locations are taking place. In contrast, spatial analyses are frequently used in studies on other decomposing substrates. For redundancy analysis, see [55] for an example with saproxylic beetles and [56] for an example with dung beetles. For an example of a polynomial generalized linear model analysis of the position with saproxylic beetles, see [57]. For an example of a spatially explicit GAMM with saproxylic beetles, see [58].

### 3.5. How to Solve Problems Related to Having a Large Amount of Systematic Variance

First, ask yourself whether all the influential variables were measured, and if the best model is being used. A research protocol excluding some influential variables as well as an inappropriate model can limit the ability of statistical tests to account for quantifiable effects. If the best model and design has been applied, my second recommendation is to accept this unexplained variance. An even better recommendation is to embrace it! It would be preposterous to hope to account for all the variance knowing the numerous factors influencing decomposition and insect colonization [59]. Most insects aggregating on and around cadavers/carcasses are hardly predictable [20,31], which contribute to this unexplained variance. Thus, this situation is not alarming and not unique to forensic entomology; the same large unexplained variance also prevails in studies of saproxylic beetles [48,49]. In the applied context of forensic entomology, it then appears more important to distinguish between what can and what cannot be explained. I firmly believe that too much time and energy has been spent researching insect species that do not have real potential for the estimation of the PMI, PIA or PCI.

## 4. Advice to Scientific Editors, Reviewers and Academic Supervisors

Only 27 out of the 160 field studies published between 1985 and 2013 in forensic anthropology, forensic entomology and forensic taphonomy had an adequate design and analysis [14] and a consultation of the recent literature shows that the situation has changed little. Therefore, it is no exaggeration to assume that most of the experimental studies in these fields that are used in court should not be admissible because they cannot generate probabilistic inference. This embarrassing situation can be explained by the fact that there is often no negative consequence related to the publication of studies using poor research methods and abusing statistical procedures [60]. While this situation is not unique to forensic entomology [61], it has far greater consequences than simply misleading colleagues and the scientific community because false-positive findings are being used to solve criminal cases. We all have a responsibility to prevent the publication of studies that fall short of experimental standards, including false preliminary studies. The prefix “a preliminary study on” has been overused in the forensic entomology literature. According to the Merriam-Webster Dictionary, “preliminary” indicates that something is a prelude to something else. However, most preliminary studies have no follow-up, this term rather serving as a loophole to allow for the publication of pseudoreplicated studies.

To avoid the publication and possible use in court of forensic studies that are poorly designed or poorly analyzed, scientific editors and reviewers should make sure that the following conditions are met by publications:*The study is devoid of experimental errors.* Scientific editors and reviewers should not be afraid to require from authors a detailed description and a map of the layout of the study. Regardless of the nature of the study, the experimental unit should always be clearly identified. To learn how to recognize the experimental unit and main experimental errors, read [12‒14,62]. Pseudoreplicated studies should never be published, even as “preliminary studies”.*If the nature of the study allows for it, an inferential statistical test that permits extrapolation of the results to case scenarios is presented*. The statistical procedures should be described in detail and an estimate of the experimental error should be evident in the tables and figures of the manuscript. If successional data are involved, the statistical test should comply with elements presented in Table 2.*If the nature of the study does not allow for it, no inferential statistical test is presented.* In a widely cited article, Hurlbert [62] suggested that good articles that refrain from using inferential statistics when these cannot be applied are worth publishing. However, the authors should explicitly recognize that the study is descriptive, thus not allowing for transposition of the results to other situations or use in court.

Inevitably, to ensure that forensic entomology studies generate probabilistic inference of quality, scientific editors and reviewers will have to step up and act as the watchdogs of the scientific method. Nobody should be reluctant to question why money, time and publication space is spent on a study that has an inappropriate design, analysis or a lack of analysis.

My last advice is for academic supervisors and concerns the training of students. To effectively understand and implement the analyzes discussed herein, forensic science students need statistical knowledge that goes far beyond classical frequentist statistics and linear models. Therefore, I urge academic supervisors to encourage their students to pursue advanced training in statistics. More than ever, as access to powerful statistical tools has become more democratized, knowledge of experimental design and statistics is proving to be an essential skill to bridge the gap between laboratory/field studies and court evidence.

## Figures and Tables

**Table 1 insects-12-00240-t001:** Issues and consequences of successional samples or measurements recovered sequentially from a decaying substrate.

Issues	Consequences
1. Data measured repetitively from a small number of sampling units and field sites	Data interdependence Autoregressive covariance structure Low statistical power Low internal validity
2. Data presenting non-linear trends	Non-linear effects Overfitting Non-Gaussian distribution
3. Datasets with a relatively large proportion of unexplained variance	High proportion of systematic variance Low external validity
4. Data affected by temporal and spatial effects	Data interrelated in time Data interrelated in space
5. Datasets including many independent and dependent variables	Autocorrelation Multicollinearity Overfitting Alpha inflation

**Table 2 insects-12-00240-t002:** Consequences and possible remedies to the issues associated with successional samples or measurements recovered sequentially from a decaying substrate.

Consequences of Successional Data	Remedies
1. Low statistical power Low internal validity Low external validity	Increase the sample size Increase the number of locations, times, and conditions
2. Data interdependence Autoregressive covariance structure Non-linear effects Non-Gaussian distribution	Generalized linear models (GLMs), generalized linear mixed models (GLMMs), generalized additive models (GAMs), generalized additive mixed models (GAMMs)
3. Autocorrelation Multicollinearity Overfitting Alpha inflation	Multivariate statistics
4. Data interrelated in time Data interrelated in space	Time series analysis Spatial statistics Repeated measures and/or spatially explicit GLMs, GLMMs, GAMs, GAMMs
5. High proportion of systematic variance	Ensure that all influential variables have been accounted for Use a model that is better suited to data Acknowledge this variability

## Data Availability

Not applicable.

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
