# Peer review of "The Pitfalls in the Path of Probabilistic Inference in Forensic Entomology: A Review"

_insects, 2021, doi:10.3390/insects12030240_

Round 1

Reviewer 1 Report

It would be unnecessary to indicate references in parentheses as  "(e.g., [number])" that often seen in this review.

For forensic entomologists with insufficient statistical knowledge, it is difficult to understand what kind of example is wrong and what kind of statistical model should be used without concrete examples. Therefore, it would be recommended to explain with actual examples as much as possible.

Author Response

Comment 1.1. It would be unnecessary to indicate references in parentheses as "(e.g., [number])" that often seen in this review.

Response to Comment 1.1: All occurrences where “(e.g., [number])” was used have been deleted.

Comment 1.2. For forensic entomologists with insufficient statistical knowledge, it is difficult to understand what kind of example is wrong and what kind of statistical model should be used without concrete examples. Therefore, it would be recommended to explain with actual examples as much as possible.

Response to Comment 1.2: When I wrote this manuscript, I made the editorial choice not to present concrete examples of errors because this is impossible without referring to the work of my colleagues, which, in my opinion, would risk turning these colleagues against me rather than making them aware of the issues discussed in the manuscript. Nonetheless, to comply with this comment, I have included additional examples from my own work and generic examples to describe effect size, Poisson distribution, spatial effects, pseudo-replication, and autocorrelation.

Reviewer 2 Report

Line 50: It might strengthen this paper to give a specific example of or definition of pseudoreplication, as that is a HUGE problem with design. 

Line 134-157: Suggest rewriting these lines, specifically clarifying the transition between "...apply them to a given case." and "If 15% of the variance...: or expanding them, because this is a challenging area for people with weak statistical backgrounds. 

Section 2.4 - in addition to the obvious temporal effects of repeated sampling of carcasses, heavy sampling of rare species or early colonizers will shift diversity and population size. Severe oversampling might also effect population-dependent effects on decomposition such as total body score.  Ken Schoenly has some work in this area. 

Line 177: taxon should be taxa, or rewrite this sentence because it's an awkward read. 

Line 183: autocorrelation may be worth defining or illustrating for reader ease. 

Line 202:  be consistent with use of commas vs. periods to indicate decimals in text vs. example equation.  

An additional suggestion - and I'm not sure exactly where it would fit - deals with the academic training necessary to effectively understand and implement GLM analysis. While I can only speak for the United States, even graduate-level statistics courses tend to only cover Gaussian distributions and linear models, with only the smallest amount of non-parametric or Bayesian analysis thrown in.  Since graduate students execute the majority of the experiemental work, and it is not feasible to add an independent statistician to every project, what recommendations would you make in terms of needed education for more rigorous outcomes. 

Author Response

Comment 2.1. Line 50: It might strengthen this paper to give a specific example of or definition of pseudoreplication, as that is a HUGE problem with design.

Response to Comment 2.1: Pseudoreplication was defined and a few examples were included.

Comment 2.2. Line 134-157: Suggest rewriting these lines, specifically clarifying the transition between "...apply them to a given case." and "If 15% of the variance...: or expanding them, because this is a challenging area for people with weak statistical backgrounds.

Response to Comment 2.2: This was rewritten.

Comment 2.3. Section 2.4 - in addition to the obvious temporal effects of repeated sampling of carcasses, heavy sampling of rare species or early colonizers will shift diversity and population size. Severe oversampling might also effect population-dependent effects on decomposition such as total body score.  Ken Schoenly has some work in this area.

Response to Comment 2.3: I searched the literature and among Ken Schoenly's publications and found no study on oversampling other than the one I published in 2013 with Jean-Philippe Michaud. I therefore refer to this study to mention oversampling at the location in the text suggested in the comment.

Comment 2.4. Line 177: taxon should be taxa, or rewrite this sentence because it's an awkward read.

Response to Comment 2.4: The sentence was rewritten.

Comment 2.5. Line 183: autocorrelation may be worth defining or illustrating for reader ease.

Response to Comment 2.5: It is now defined, and an example was included.

Comment 2.6. Line 202:  be consistent with use of commas vs. periods to indicate decimals in text vs. example equation. 

Response to Comment 2.6: This was a mistake. Thank you!

Comment 2.7. An additional suggestion - and I'm not sure exactly where it would fit - deals with the academic training necessary to effectively understand and implement GLM analysis. While I can only speak for the United States, even graduate-level statistics courses tend to only cover Gaussian distributions and linear models, with only the smallest amount of non-parametric or Bayesian analysis thrown in.  Since graduate students execute the majority of the experiemental work, and it is not feasible to add an independent statistician to every project, what recommendations would you make in terms of needed education for more rigorous outcomes.

Response to Comment 2.7: I want to thank the reviewer for this excellent suggestion. The last paragraph of the discussion was rewritten to discuss this point.

Reviewer 3 Report

This manuscript is well written, and addresses long-standing issues in forensic entomology on the use of statistical analysis.  A few points to consider are below:

Line 79:  overlooks should be overlook

Table 1:  The spacing for the text has the column margins too close together.  The space should be increased so the columns are distinct.

Table 2:  See above comment on table 1.

Line 275:  Insert “if” so the sentence reads “if the best model is being used.”

Author Response

Comment 3.1. This manuscript is well written, and addresses long-standing issues in forensic entomology on the use of statistical analysis.

Response to Comment 3.1: I’d like to thank the reviewer for this comment. To be honest, I think I should have written this text a long time ago.

Comment 3.2. Line 79:  overlooks should be overlook

Response to Comment 3.2: This was modified as suggested.

Comment 3.3. Table 1:  The spacing for the text has the column margins too close together.  The space should be increased so the columns are distinct.

Response to Comment 3.3: The spacing was increased.

Comment 3.4. Table 2:  See above comment on table 1.

Response to Comment 3.4: The spacing was increased.

Comment 3.5. Line 275:  Insert “if” so the sentence reads “if the best model is being used.”

Response to Comment 3.5: This was modified as suggested.
